# Adaptively Robust and Sparse $K$-means Clustering

**Hao Li**
*Graduate School of Economics*
*Keio University*

*lee1995hao@keio.jp*

**Shonosuke Sugasawa**
*Faculty of Economics*
*Keio University*

*sugasawa@econ.keio.ac.jp*

**Shota Katayama**
*Faculty of Economics*
*Keio University*

*katayama@econ.keio.ac.jp*

**Reviewed on OpenReview:** *https://openreview.net/forum?id=EhC84fT2yA*

## Abstract

While $K$-means is known to be a standard clustering algorithm, its performance may be compromised due to the presence of outliers and high-dimensional noisy variables. This paper proposes adaptively robust and sparse $K$-means clustering (ARSK) to address these practical limitations of the standard $K$-means algorithm. For robustness, we introduce a redundant error component for each observation, and this additional parameter is penalized using a group sparse penalty. To accommodate the impact of high-dimensional noisy variables, the objective function is modified by incorporating weights and implementing a penalty to control the sparsity of the weight vector. The tuning parameters to control the robustness and sparsity are selected by Gap statistics. Through simulation experiments and real data analysis, we demonstrate the proposed method's superiority to existing algorithms in identifying clusters without outliers and informative variables simultaneously.

## 1 Introduction

Identifying clustering structures is recognized as an important task in pursuing potential heterogeneity in datasets. While there are a variety of clustering methods, $K$-means clustering (Forgy, 1965) is the most standard clustering algorithm and is widely used in many scientific applications. However, real-world datasets present significant difficulties, such as the presence of outliers and high-dimensional noisy variables. Regarding the outlier issues, there are some attempts to robustify the standard $K$-means such as trimmed $K$-means (Cuesta-Albertos et al., 1997) and robust $K$-means (Klochkov et al., 2021). On the other hand, high-dimensional noisy characteristics are typically addressed through clustering approaches that focus on sparsity for variable selection. In particular, Witten & Tibshirani (2010) proposed a lasso-type penalty to the variable weights incorporated in the clustering objective function. However, the efficacy of this algorithm might be lost if the dataset contains a significant number of outliers. While several methods address either of the two aspects (the existence of outliers and noisy variables), practically useful methods to address both aspects simultaneously are scarce. One exception is the method proposed by Kondo et al. (2016). However, this approach requires the assumption of a trimming level in advance. In contrast, Brodinová et al. (2019) introduces an approach beginning with eliminating outliers identified by the model. However, this methodology potentially leads to reduced sample size and may result in information loss. Moreover, the excessive variations in gradient magnitudes may destabilize when optimizing the variable weight process unless all outliers are eliminated in advance.

In addition to clustering methods based on objective functions, probabilistic clustering using mixture models is also popular. It is typically done by fitting multivariate Gaussian mixtures. While several approaches

have been proposed for stable estimation and clustering under existence of outliers (e.g. Coretto & Hennig, 2016; Fujisawa & Eguchi, 2006; Punzo & McNicholas, 2016; Sugasawa & Kobayashi, 2022; Yang et al., 2012), such approach suffers from a huge number of parameters (especially for modelling covariance matrix) to be estimated when the dimension is large. Therefore, these methods would not be a reasonable solution when our primary focus is on clustering.

To address the difficulty of a series of sparse trimming-cluster and probabilistic-cluster approaches, we propose a new approach based on regularization techniques for sparsity as well as robustness, called adaptively robust and sparse $K$-means (ARSK). We introduce an error component for each observation as an additional parameter to absorb the undesirable effects caused by outliers, making the clustering algorithm robust. We employ a group lasso penalty and a group smoothly clipped absolute deviation (SCAD) penalty (Wang et al., 2007) to estimate these parameters during clustering steps. Meanwhile, to reduce the data dimension, we also introduce a SCAD penalty (Fan & Li, 2001) to weights for each variable to eliminate noise variables irrelevant to clustering. Based on our analysis of real-world datasets, in the ultra-high dimensional setting (i.e., the dataset dimension exceeding 700), the result procured through the implementation of the lasso penalty proposed by Witten & Tibshirani (2010) frequently poor performance. Therefore, we have developed a methodology based on the SCAD penalty. The strong sparsity property of the SCAD penalty effectively improves the selection of relevant variables in the ultra-high dimensional scenario. We then develop an efficient computation algorithm to minimize the proposed objective function. While the proposed method has two tuning parameters controlling robustness (the number of outliers to be detected) and sparsity (the level of the sparsity of the variable weight), we adopt a modified version of the Gap statistics (Tibshirani et al., 2001) to determine the optimal values of them. Hence, the level of robustness and sparsity is adaptively determined in a fully data-dependent manner.

The paper is organized as follows. Section 2 reviews the $K$-means algorithm and its extended model introduces the proposed method ARSK and presents its details and pseudocode. We also present the selection approach of tuning parameters and its pseudocode. while Section 3 presents the tuning parameter selection result. Both two algorithms are applied to various scenarios of artificial datasets and several real-world datasets. This paper ends in Section 4 with concluding remarks. R code implementing the proposed method is available at the GitHub repository (`https://github.com/lee1995hao/ARSK`).

## 2 Robust and Sparse $K$-means Clustering

### 2.1 Conventional $K$-means algorithm

Suppose that we observe a $p$-dimensional vector, $\boldsymbol{X}_{i,:} = (X_{i1}, \ldots, X_{ip})$, for $i = 1, \ldots, n$, and that we are interested in clustering $\boldsymbol{X}_{i,:}$. Given the number of clusters, $K$, the conventional $K$-means algorithm is to find the optimal clustering by minimizing the following objective function:

$$\min_{c_1, \ldots, c_K} \sum_{k=1}^{K} \frac{1}{n_k} \sum_{i, i' \in c_k} \|\boldsymbol{X}_{i,:} - \boldsymbol{X}_{i',:}\|^2, \tag{1}$$

where $\|\boldsymbol{X}_{i,:} - \boldsymbol{X}_{i',:}\|^2$ is the squared value of $L_2$-distance between two observed vectors. Here $c_k$ is a set of indices representing a cluster of observations and $n_k$ is the size of $c_k$, such that $\cup_{k=1}^{K} c_k = \{1, \ldots, n\}$ and $\sum_{k=1}^{K} n_k = n$, where $c_k \cap c_{k'} = \phi$ for $k \neq k'$. The objective function (1) can be easily optimized by iteratively updating cluster means and assignment (e.g. Lloyd, 1982).

As demonstrated in Witten & Tibshirani (2010), the objective function (1) can be reformulated in terms of the between-cluster sum of squares, given by $\max_{c_1, \ldots, c_k} \sum_{j=1}^{p} Q_j(\boldsymbol{X}_{:,j}; \Theta)$, where

$$Q_j(\boldsymbol{X}_{:,j}; \Theta) = \frac{1}{n} \sum_{i=1}^{n} \sum_{i'=1}^{n} (X_{ij} - X_{i'j})^2 - \sum_{k=1}^{K} \frac{1}{n_k} \sum_{i, i' \in c_k} (X_{ij} - X_{i'j})^2, \tag{2}$$

with $\boldsymbol{X}_{:,j} = (X_{1j}, \ldots, X_{nj})$ and $\Theta$ being a collection of $c_1, \ldots, c_K$. Witten & Tibshirani (2010) employed the modified version of the objective function (2), given by $\sum_{j=1}^{p} \omega_j Q_j(\boldsymbol{X}_{:,j}; \Theta)$ with an appropriate constrained on $\omega_j$, for conducting variable selection and clustering simultaneously.

## 2.2 Robust and sparse $K$-means algorithm

The objective function (2) is sensitive to outliers since the approach relies on the $L_2$ distance between each observation. As a result, even a small quantity of deviating observations can affect the $K$-means algorithm. In order to robustify the $K$-means approach, we modify the function $Q_j(\boldsymbol{X}_{:,j}; \Theta)$ as follows:

$$
\begin{aligned}
Q_j^R(\boldsymbol{X}_{:,j}; \Theta, \boldsymbol{E}) = &\frac{1}{n} \sum_{i=1}^{n} \sum_{i'=1}^{n} \{(X_{ij} - E_{ij}) - (X_{i'j} - E_{i'j})\}^2 \\
&- \sum_{k=1}^{K} \frac{1}{n_k} \sum_{i,i' \in c_k} \{(X_{ij} - E_{ij}) - (X_{i'j} - E_{i'j})\}^2,
\end{aligned}
\tag{3}
$$

where $\boldsymbol{E}$ is an $n \times p$ error matrix, which is a collection of $E_{ij}$ and $E_{ij}$ is an additional location parameter for each $X_{ij}$ such that $E_{ij}$ will be non-zero values for outlying observations $X_{ij}$ to prevent an undesirable effects from outliers. While the number of elements of $\boldsymbol{E}$ is the same as the number of observations, it can be assumed that $\boldsymbol{E}$ is sparse in the sense that most elements of $\boldsymbol{E}$ are 0. That is, most elements are not outliers. To stably estimate $\boldsymbol{E}$, we consider penalized estimation for $E_{ij}$ when optimizing the function (3). Such approaches are used in the robust fitting of regression models (e.g. She & Owen, 2011; Katayama & Fujisawa, 2017).

Based on the robust objective function (3), we propose an objective function for robust and sparse $K$-means algorithm as follows:

$$
L(\Theta, \boldsymbol{E}, \boldsymbol{w}; \lambda) = \sum_{j=1}^{p} w_j Q_j^R(\boldsymbol{X}_{:,j}; \Theta, \boldsymbol{E}_{:,j}) - \sum_{i=1}^{n} P_1(\|\boldsymbol{E}_{i,:}\|_2; \lambda_1) - \sum_{j=1}^{p} (P_2(w_j; \lambda_2) + \frac{1}{2} w_j^2),
\tag{4}
$$

where $P_1$ and $P_2$ are group penalty functions and penalty functions for sparse (shrinkage) estimation of $\boldsymbol{E}_{i,:}$ and $w_j$, respectively, and $\lambda = (\lambda_1, \lambda_2)$ is a set of tuning parameters. To find a unique solution for the weight coefficients $w_j$, we add a quadratic term $\frac{1}{2} \sum_{j=1}^{p} w_j^2$ to the objective function. Then, the robust and sparse $K$-means clustering can be defined as $\widehat{\Theta}$ such that

$$
(\widehat{\Theta}, \widehat{\boldsymbol{E}}, \widehat{\boldsymbol{w}}) = \mathrm{argmax}_{\Theta, \boldsymbol{E}, \boldsymbol{w} \in \mathcal{H}} L(\Theta, \boldsymbol{E}, \boldsymbol{w}; \lambda),
$$

$$
\mathcal{H} = \left\{ w_j \in \mathbb{R}^+ : \sqrt{\sum_{j=1}^{p} w_j^2} = 1 \right\}.
$$

In what follows, we employ two specific forms of group penalty functions $P_1(\|\boldsymbol{E}_{i,:}\|_2; \lambda_1)$: one type is the convex group penalty function, which is the group lasso penalty as $\lambda_1 \|\boldsymbol{E}_{i,:}\|_2$ (Yuan & Lin, 2006), and the other type is the nonconvex group penalty function, which is the group SCAD penalty (Huang et al., 2012) as

$$
P_{\lambda_1}^{SCAD}(\|\boldsymbol{E}_{i,:}\|_2) = \begin{cases} \lambda_1 \|\boldsymbol{E}_{i,:}\|_2 & \text{if } \|\boldsymbol{E}_{i,:}\|_2 \le \lambda_1; \\ -(\|\boldsymbol{E}_{i,:}\|_2^2 - 2a\lambda_1 \|\boldsymbol{E}_{i,:}\|_2 + \lambda_1^2)/2(a-1) & \text{if } \lambda_1 < \|\boldsymbol{E}_{i,:}\|_2 \le a\lambda_1; \\ (a+1)\lambda_1^2/2 & \text{if } \|\boldsymbol{E}_{i,:}\|_2 > a\lambda_1. \end{cases}
$$

We also consider both convex and nonconvex penalties for the penalty term $P_2(w_j; \lambda_2)$: the convex standard lasso penalty $P_2(w_j; \lambda_2) = \lambda_2 |w_j|$ and, the non-convex SCAD penalty expressed as $P_2(w_j; \lambda_2) = P_{\lambda_2}^{SCAD}(w_j)$, where $P_{\lambda_2}^{SCAD}(w_j)$ define as:

$$
P_{\lambda_2}^{SCAD}(w_j) = \begin{cases} \lambda_2 |w_j| & \text{if } |w_j| \le \lambda_2; \\ -(|w_j|^2 - 2a\lambda_2 |w_j| + \lambda_2^2)/2(a-1) & \text{if } \lambda_2 < |w_j| \le a\lambda_2; \\ (a+1)\lambda_2^2/2 & \text{if } |w_j| > a\lambda_2. \end{cases}
$$

When applying the $L_1$ or SCAD penalties, we seek the optimal weight vector of $\boldsymbol{w} = (w_1, \ldots, w_p)$, which is normalized to have a unit $L_2$-norm, i.e., $\sqrt{\sum_{j=1}^p w_j^2} = 1$. This approach helps to promote a more balanced solution (Zou & Hastie, 2010).

## 2.3 Optimization algorithm

The block-coordinated decent method can optimise the proposed objective function (4) for updating parameters. Regarding the update of the clustering assignment $\Theta$, we note that the maximization of $L(\Theta, \boldsymbol{E}, \boldsymbol{w}; \lambda)$ with respect to $\Theta$ given $\boldsymbol{E}$ and $\boldsymbol{w}$ is equivalent to minimizing

$$L(\Theta|\boldsymbol{E}, \boldsymbol{w}; \lambda) = \sum_{j=1}^p w_j \sum_{k=1}^K \frac{1}{n_k} \sum_{i,i' \in c_k} \left\{ (X_{ij} - E_{ij}) - (X_{i'j} - E_{i'j}) \right\}^2.$$

The optimal solution to this equation is achieved by assigning each adjusted observation point to the sample mean of the adjusted observation with the smallest squared Euclidean distance (e.g. Lloyd, 1982), given by

$$\text{argmin}_\Theta \sum_{j=1}^p \sum_{k=1}^K \sum_{i \in c_k} w_j \left\{ X_{ij} - E_{ij} - \frac{1}{n_k} \sum_{i \in c_k} (X_{ij} - E_{ij}) \right\}^2, \tag{5}$$

which can be easily computed by using the standard $K$-means algorithm.

Given clustering assignment $\Theta$ and variable weight $\boldsymbol{w}$, the error matrix $\boldsymbol{E}$ can be updated as

$$\text{argmin}_E \left\{ \frac{1}{2} \sum_{j=1}^p w_j \sum_{k=1}^K \sum_{i \in c_k} \left( X_{ij} - E_{ij}^* - \mu_{kj}^* \right)^2 + \sum_{i=1}^n P_1(\|\boldsymbol{E}_{i,:}\|_2; \lambda_1) \right\}, \tag{6}$$

where $\mu_{kj}^* = n_k^{-1} \sum_{i \in k} (X_{ij} - E_{ij}^*)$ and $E_{ij}^*$ is the optimized value of $E_{ij}$ from the previous iteration.

As previously discussed, the variable $\boldsymbol{E}$ plays a crucial role in robustness. If an observation does not fall into any cluster, vector $\boldsymbol{E}_{i,:}$ elements will exhibit large values. The penalty $P_1(\|\boldsymbol{E}_{i,:}\|_2; \lambda_1)$ controls the robustness of the model. We also note that under $\lambda_1 \to \infty$, all $\boldsymbol{E}_{i,:}$ equal a $p$-dimensional zero vector. As a result, the objective function (6) will be equivalent to traditional $K$-means clustering.

In the minimization problem (6), as previously mentioned, we employ the group lasso penalty and group SCAD penalty (Tibshirani, 1996; Antoniadis & Fan, 2001). The optimizers are obtained by applying the thresholding function corresponding to the group penalty to $\boldsymbol{X}_{i,:} - \boldsymbol{\mu}_{k,:}$. These two types of group penalty functions correspond to two distinct categories of thresholding functions: the group lasso penalty function corresponds to the multivariate soft-threshold operator, where $\boldsymbol{E}_{i,:}$ can be calculated through

$$\boldsymbol{E}_{i,:} = (\boldsymbol{X}_{i,:} - \boldsymbol{\mu}_{k,:}) \max \left( 0, 1 - \frac{\lambda_1}{\|\boldsymbol{X}_{i,:} - \boldsymbol{\mu}_{k,:}\|_2} \right) \tag{7}$$

as proposed by Witten (2013), while the group SCAD penalty function corresponds to the multivariate SCAD-threshold operator introduced by Huang et al. (2012), which $\boldsymbol{E}_{i,:}$ can be calculated as:

$$\boldsymbol{E}_{i,:} = \begin{cases} \boldsymbol{S}(\boldsymbol{z}_i; \lambda_1), & \text{if } \|\boldsymbol{z}_i\|_2 \leq 2\lambda_1, \\ (a-2)^{-1}(a-1)\boldsymbol{S}(\boldsymbol{z}_i; (a-1)^{-1}a\lambda_1), & \text{if } 2\lambda_1 < \|\boldsymbol{z}_i\|_2 \leq a\lambda_1, \\ \boldsymbol{z}_i, & \text{if } \|\boldsymbol{z}_i\|_2 > a\lambda_1. \end{cases} \tag{8}$$

where $\boldsymbol{z}_i = \boldsymbol{X}_{i,:} - \boldsymbol{\mu}_{k,:}$ and $\boldsymbol{S}(\boldsymbol{z}_i; \lambda)$ represents multivariate soft-threshold operator. In this paper, we set $a$ as equal to 3.7, as recommended in Fan & Li (2001).

In addition to the previously mentioned multivariate soft-threshold operator and multivariate SCAD-threshold operator, The application of the aforementioned methods can also be extended to the group

minimax concave penalty (Huang & Zhang, 2010) associated with the multivariate minimax concave penalty-threshold operator. However, due to its similarity to the SCAD penalty, we exclusively employed the SCAD penalty. Moreover, we suggest avoiding applying the group hard penalty (Antoniadis, 1996), as optimizing it is inherently difficult. Furthermore, during our simulation process, we discovered that it is easy to converge to local minima.

As discussed by Witten (2013) and She & Owen (2011), this formulation establishes a connection between the framework of robust M-estimation and the proposed model (6) with clustering centre $\mu_{kj}^*$ for each $k$ held fixed.

As described above, the matrix of errors acquired from the algorithm is weighted based on the current weights. Therefore, it is necessary to restore the weighted error matrix $\boldsymbol{E}$ to its original unweighted state by dividing the current weight before we go to the next step to calculate the new weight for each variable (if the weight of a certain variable is zero, then $\boldsymbol{E}_{:,j}$ will divide one instead of zero).

Finally, given $\boldsymbol{E}$ and $\Theta$, the minimization of $L(\Theta, \boldsymbol{E}, \boldsymbol{w}; \lambda)$ with respect to $\boldsymbol{w}$ is equivalent to

$$\text{argmax}_{\boldsymbol{w} \in \mathcal{H}} \left\{ \sum_{j=1}^{p} w_j Q_j^R(\boldsymbol{X}_{:,j}; \Theta, \boldsymbol{E}_{:,j}) - \sum_{j=1}^{p} \left( P_2(w_j; \lambda_2) + \frac{1}{2} w_j^2 \right) \right\}, \tag{9}$$

where $\mathcal{H} = \left\{ w_j \in \mathbb{R}^+ : \sqrt{\sum_{j=1}^{p} w_j^2} = 1 \right\}$. We show in the Appendix that the final solution to (9) can be denoted as

$$w_j = \frac{S(Q_j^R(\boldsymbol{X}_{:,j}; \Theta, \boldsymbol{E}_{:,j}); \lambda_2)}{\sqrt{\sum_{j=1}^{p} (S(Q_j^R(\boldsymbol{X}_{:,j}; \Theta, \boldsymbol{E}_{:,j}); \lambda_2))^2}}.$$

When applying the $P_2(w_j; \lambda_2) = \lambda_2 |w_j|$ penalty in (9), $S$ can be represented by the soft-thresholding operator as

$$S(x; \lambda_2) = \begin{cases} x - \lambda_2, & \text{if } x > \lambda_2 \\ 0, & \text{if } |x| \leq \lambda_2 \\ x + \lambda_2, & \text{if } x < -\lambda_2 \end{cases}$$

When applying the $P_2(w_j; \lambda_2) = P_{\lambda_2}^{SCAD}(w_j)$ penalty in (9), $S$ can be represented by the SCAD-thresholding operator as

$$S(x; \lambda_2) = \begin{cases} \text{sgn}(x)(|x| - \lambda_2)_+ & \text{if } |x| \leq 2\lambda_2 \\ \{(a - 1)x - a\lambda_2 \cdot \text{sign}(x)\} / (a - 2) & \text{if } 2\lambda_2 \leq |x| < a\lambda_2 \\ x & \text{if } a\lambda_2 \leq |x| \end{cases}$$

The parameter $\lambda_2$ has a crucial role in controlling the sparsity in the variable weight vector. A higher value of $\lambda_2$ results in increased sparsity of the variable vector $\boldsymbol{w}$. Specifically, a higher weight value $w_j$ indicates greater importance of the $j$th variable in the clustering process. Conversely, when $w_j$ is equal to zero, it signifies that the $j$th variable does not contribute to the clustering.

In order to gain a deeper comprehension of the algorithmic process we have proposed, we summarize the aforementioned procedures in a pseudocode in Algorithm 1. Moreover, the computation complexity of $T$ iteration of Algorithm 1 can be represented $O(nptTK)$, assuming that the number of iterations in the 4th step is $O(t)$.

## 2.4 Adaptation of the tuning parameters by the robust Gap statistics

There are two tuning parameters in the proposed algorithm: $\lambda_1$ and $\lambda_2$. The tuning parameter $\lambda_1$ is crucial for finding the error matrix and impacts the detection of outliers, and the tuning parameter $\lambda_2$ plays a crucial role in filtering out variables contributing to the clustering process. We here provide the robust Gap statistics to determine these parameters.

---

**Algorithm 1** : Iterative algorithm for ARSK

---

1: Initialize the weight vector $\boldsymbol{w}$ as $w_j = 1/\sqrt{p}$, set tolerance $\varepsilon > 0$ and $r = 0$.

2: Initialize the error matrix by setting $E_{ij}^{(0)}$ of the 80% of data farthest from all data center points to $X_{ij}$; the others set as 0.

3: **repeat**

4:  Run the clustering algorithm on the weighted dataset $X_{ij}^* = w_j^{(r)}(X_{ij} - E_{ij}^{(r)})$ by

$$\max_{c_1,\dots,c_k} \left\{ \sum_{j=1}^p \left( \left( X_{ij}^* - \frac{1}{n}\sum_{i=1}^n X_{ij}^* \right)^2 - \sum_{k=1}^K \sum_{i \in c_k} \left( X_{ij}^* - \frac{1}{|n_k|}\sum_{i \in c_k} X_{ij}^* \right)^2 \right) \right\}$$

5:  Calculate the new $E_{ij}$ by (7) or (8), which incorporates the cluster-specific means $\boldsymbol{\mu}_{k,:} = |n_k|^{-1} \sum_j \sum_{i \in c_k} X_{ij}^*$

6:  Until the (6) converges, we can obtain the error matrix $E_{ij}^{(r+1)}$ and the cluster arrangement $\Theta^{(r+1)}$. Keep those results for the next variable selection phase.

7:  Restore the error matrix by setting:

$$E_{ij}^{(r+1)} := E_{ij}^{(r+1)}/\sqrt{w_j^{(r)}}$$

8:  Arrange the cluster $\Theta^{(r+1)}$ and compute the $Q_j^R(\boldsymbol{X}_{:,j}; \Theta^{(r+1)}, \boldsymbol{E})$ for different variables by:

$$Q_j^R(\boldsymbol{X}_{:,j}; \Theta^{(r+1)}, \boldsymbol{E}) = \sum_{i=1}^n \left( X_{ij}' - \frac{1}{n}\sum_{i=1}^n X_{ij}' \right)^2 - \sum_{k=1}^K \sum_{i \in c_k^{(r+1)}} \left( X_{ij}' - \frac{1}{|n_k|}\sum_{i \in c_k^{(r+1)}} X_{ij}' \right)^2,$$

 where $X_{ij}' = X_{ij} - E_{ij}^{(r+1)}$.

9:  Compute new variable weight by:

$$w_j^{(r+1)} = \frac{S(Q_j^R(\boldsymbol{X}_{:,j}; \Theta, \boldsymbol{E}_{:,j}); \lambda_2)}{\sqrt{\sum_{j=1}^p (S(Q_j^R(\boldsymbol{X}_{:,j}; \Theta, \boldsymbol{E}_{:,j}); \lambda_2))^2}}$$

10:  $r = r + 1$

11: **until** convergence of the criterion $(\sum_j |w_j^{(r)}|)^{-1} \sum_j |w_j^{(r+1)} - w_j^{(r)}| < \varepsilon$ is met.

12: **Output**: $\boldsymbol{w}$, $\boldsymbol{E}$, $\Theta$

---

The Gap statistics were originally proposed in Tibshirani et al. (2001) for selecting the number of clusters in the $K$-means algorithm. Note that the Gap statistics is constructed as a function of the between-cluster sum of squares, $D = \sum_{j=1}^p w_j(n^{-1}\sum_{i=1}^n \sum_{i'=1}^n (X_{ij} - X_{i'j})^2 - \sum_{k=1}^K n_k^{-1}\sum_{i,i' \in c_k}(X_{ij} - X_{i'j})^2)$, for the original dataset. However, since $D$ is sensitive to outliers, it is not suitable for selecting the tuning parameters under existence of outliers. Hence, we propose a robust version of Gap statistics by adding the error part to each observation when calculating the $D_{\lambda_2,\lambda_1}^R$. As we mentioned before, the error part can outweigh the influence of the outlier, so that we define $\text{Gap}_{\lambda_2,\lambda_1}$ as

$$\text{Gap}_{\lambda_2,\lambda_1} = \log(D_{\lambda_2,\lambda_1}^R) - \frac{1}{B}\sum_{b=1}^B \log(D_{\lambda_2,\lambda_1}^{R(b)}), \quad D_{\lambda_2,\lambda_1}^R = \sum_{j=1}^p w_j Q_j^R(\boldsymbol{X}_{:,j}; \Theta, \boldsymbol{E}_{:,j}), \tag{10}$$

and the optimal value of $\lambda_2$ and $\lambda_1$ corresponding to the largest value of $\text{Gap}_{\lambda_2,\lambda_1}$. Note that $D_{\lambda_2,\lambda_1}^R$ is a weighted robust between-cluster sum of squares defined in (3), and $D_{\lambda_2,\lambda_1}^{R(b)}$ is a version with $b$th permuted datasets. Here, $B$ denotes the number of permuted datasets generated by randomly selecting from the original dataset. Increasing the number of permuted datasets improves the accuracy of selecting the true parameters. The between-cluster sum of squares, taking account of the variable weight, is adopted in Witten & Tibshirani (2010), so our version used in (10) can be regarded as its robust extension.

Since it may be computationally intensive to search the optimal value of $(\lambda_2, \lambda_1)$ by a grid search method, we conduct an alternating optimization algorithm for tuning parameter search. Specifically, we first set a suitable value $\lambda_1^{\dagger}$ for $\lambda_1$ and compute the optimal value $\lambda_2^*$ of $\lambda_2$ by maximizing $\mathrm{Gap}_{\lambda_2, \lambda_1^{\dagger}}$. Then, we obtain the optimal value $\lambda_1^*$ of $\lambda_1$ by maximizing $\mathrm{Gap}_{\lambda_2^*, \lambda_1}$. Using this search algorithm instead of the grid search method can save a significant computation time. We also offer a pseudocode for this procedure for easy understanding of this search algorithm given in Algorithm 2.

---

**Algorithm 2** : Selection of $(\lambda_2, \lambda_1)$ by the robust Gap statistics

---

1: Set some value for $\lambda_1^{\dagger}$ and maximize the following Gap statistics with respect to $\lambda_2$:

$$\mathrm{Gap}_{\lambda_2} = \log(D^R_{\lambda_2, \lambda_1^{\dagger}}) - \frac{1}{B} \sum_{b=1}^{B} \log(D^{R(b)}_{\lambda_2, \lambda_1^{\dagger}})$$

to obtain the optimal $\lambda_2^*$.

2: Fix $\lambda_2 = \lambda_2^*$ and optimize the following Gap statistics with respect to $\lambda_1$:

$$\mathrm{Gap}_{\lambda_1} = \log(D^R_{\lambda_2^*, \lambda_1}) - \frac{1}{B} \sum_{b=1}^{B} \log(D^{R(b)}_{\lambda_2^*, \lambda_1})$$

to obtain the optimal $\lambda_1^*$.

3: **Output**: the optimal set of tuning parameters, $(\lambda_2^*, \lambda_1^*)$

---

## 3 Numerical Studies

### 3.1 Experimental setup

In this section, we explore the ability of the proposed clustering method. We consider that each observation $\boldsymbol{x}_i$ is generated independently from a multivariate normal distribution, given that the observation belongs to cluster $k$. Specifically, for an observation $\boldsymbol{x}_i$ in cluster $k$, we have $\boldsymbol{x}_i \sim \mathcal{N}(\boldsymbol{\mu}_{k,:}, \boldsymbol{\Sigma}_p)$, where $\boldsymbol{\mu}_{k,:} \in \mathbb{R}^p$ denotes the mean vector for cluster $k$, and $\boldsymbol{\Sigma}_p \in \mathbb{R}^{p \times p}$ represents the covariance matrix. Each element of $\boldsymbol{\mu}_{k,:}$ is independently sampled from from either $U(-6, -3)$ or $U(3, 6)$. To simulate scenarios involving outliers, we introduce a contaminated error distribution with a multivariate normal mixture, represented as $(1 - \pi)\mathcal{N}(\boldsymbol{\mu}_{k,:}, \boldsymbol{\Sigma}_p) + \pi\mathcal{N}(\boldsymbol{\mu}_{k,:} + b_j, \boldsymbol{\Sigma}_p)$ where $j \in (1, \ldots, p)$ and $\pi \in [0, 1]$ represents the proportion of outliers for each cluster. In the simulation study, we consider two types of $\boldsymbol{\Sigma}_p$. In the first scenario, the assumption is made that all variables are independent. In another scenario, we assume there is a correlation structure among variables. To introduce randomness to the appearance of outliers, $b_j$ is generated with a random number from one of two uniform distributions: $U(-13, -7)$ or $U(7, 13)$. In this setup, we assume that some explanatory variables are significantly associated with the response, while the remaining variables are redundant for variable selection. To mimic this situation, we randomly set $q$ elements of $\boldsymbol{\mu}_{k,:}$ to non-zero values while the remaining elements are set to zero. Consequently, a successful method should accurately identify the $q$ positions where the weights $w_j$ corresponding to the true significant variables are non-zero while ensuring that the weights $w_j$ of the remaining variables are set to zero.

In order to evaluate the approach's clustering accuracy capability, we employ the clustering error rate (CER) (Rand, 1971). The CER measures the extent to which the model's predicted partition $\hat{C}$ for a set of $n$ observations is consistent with the true labels $C$.

The CER is defined as

$$\mathrm{CER}(\hat{C}, C) = \binom{n}{2}^{-1} \sum_{i < i'} \left| 1_{\hat{C}_{(i,i')}} - 1_{C_{(i,i')}} \right|, \tag{11}$$

where, $1_{C_{(i,i')}}$ and $1_{\hat{C}_{(i,i')}}$ indicate whether the $i$th and $i'$th observations belong to the same cluster. Lower CER indicates better performance in clustering accuracy or outlier detection.

Furthermore, in order to evaluate each method's proficiency in variable sparsity, we apply two criteria for variable selection. The variable true positive rate (TPR) indicates the approach's success in finding informative variables. The true negative rate (TNR) represents the successful identification of non-informative variables. If both criteria are closer to 1, it indicates that the model provides a more comprehensive explanation of the structure of the variables predicted in the study.

### 3.2 Simulation 1: selection of turning parameter

In this subsection explores the new Gap statistics' capability to directly search for the tuning parameters of robustness and variable vector sparsity using Algorithm 2. Initially, we considered 3 clusters, each containing 50 observations, number of variables as $p = 50$, with $q = 5$, and all variables are independent, i.e., $\mathbf{\Sigma}_p = \mathbf{I}_p$. A proper of hyperparameters $\lambda_1$ and $\lambda_2$ should make the model accurately identify the structure of the clustering data in different contamination levels for $\pi$ in $\{0, 0.1, 0.2, 0.3\}$. The proposed tuning parameter search strategy generates both the robustness parameter $\lambda_1$ and the variable selection tuning parameter $\lambda_2$ using exponential decay. We consider the number of permuted datasets to be 25 (i.e., $B = 25$). We employed different thresholding function types, where the first thresholding function is used to control model sparsity, and the second thresholding function is used to control model robustness. The tuning parameter search process is repeated 30 times for 30 different datasets, and the resulting table is presented in Table 3.2.

| threshold type | contamination level | number of detected outliers | number of detected informative variable |
|---|---|---|---|
| soft-soft | $\pi = 0$ | 0 (0) | 4.300 (0.000) |
| | $\pi = 0.1$ | 15.56 (6.425) | 4.533 (0.618) |
| | $\pi = 0.2$ | 28.73 (2.434) | 4.200 (0.871) |
| | $\pi = 0.3$ | 33.10 (14.76) | 13.60 (7.203) |
| soft-SCAD | $\pi = 0$ | 0 (0) | 4.400 (0.000) |
| | $\pi = 0.1$ | 14.43 (0.495) | 6.310 (1.854) |
| | $\pi = 0.2$ | 28.68 (1.349) | 4.566 (0.495) |
| | $\pi = 0.3$ | 42.36 (2.287) | 6.800 (2.946) |
| SCAD-soft | $\pi = 0$ | 0 (0) | 4.545 (0.687) |
| | $\pi = 0.1$ | 14.30 (0.483) | 5.400 (0.469) |
| | $\pi = 0.2$ | 27.11 (6.166) | 5.941 (1.748) |
| | $\pi = 0.3$ | 38.63 (16.63) | 9.272 (4.540) |
| SCAD-SCAD | $\pi = 0$ | 0.090 (0.333) | 4.272 (0.881) |
| | $\pi = 0.1$ | 13.33 (1.258) | 5.833 (3.125) |
| | $\pi = 0.2$ | 24.10 (5.065) | 4.500 (2.068) |
| | $\pi = 0.3$ | 44.00 (4.2110) | 7.187 (1.223) |

Table 1: Using the robust Gap statistic to select the optimal tuning parameters for the RSKC algorithm based on soft-soft-thresholding, soft-SCAD-thresholding, SCAD-soft-thresholding, and SCAD-SCAD-thresholding, we report the average and its standard error (in parentheses) of the number of detected outliers and the number of detected informative variables.

Table 3.2 summarizes the results of the evaluation measures. Overall, a good tuning parameter $\lambda_1$ should accurately identify the number of outliers under different contamination levels. As the contamination level $\pi$ increases from 0 to 0.3, we observe that the four thresholding methods demonstrate varying outlier detection capabilities across different contamination levels. At $\pi = 0.1$, the detected number of outliers ranges from 13.33 to 15.56, with soft-soft-thresholding achieving the highest average of 15.56 (standard error: 6.425). When $\pi$ increases to 0.2, the detected number of outliers falls between 24.10 and 28.73, with soft-soft and soft-SCAD-thresholding showing similar performance. This demonstrates that the Gap statistics can help the ARKC method determine $\lambda_1$ to some extent. However, when the contamination level reaches 0.3, the soft-soft-thresholding ARSK method detects 33.10 outliers on average, which is lower than the SCAD-SCAD-thresholding ARSK method (44.00 outliers) and deviates from the true number of outliers.

Regarding the selection of the tuning parameter $\lambda_2$, a good tuning parameter should accurately identify the number of informative variables under different contamination levels. When the data is clean ($\pi = 0$),

all four methods identify around 4 to 5 informative variables, which is close to 5. As the contamination level increases, the number of detected informative variables generally increases, with some fluctuations. At $\pi = 0.3$, the soft-soft-thresholding ARSK method detects 13.60 informative variables on average, which significantly deviates from the 5. In contrast, the SCAD-SCAD-thresholding ARSK method detects 7.187 informative variables, showing better performance in the presence of high contamination. Furthermore, errors in identifying informative variables can lead to significant biases in outlier detection.

Considering these results, we can conclude that the proposed parameter selection algorithm based on the Gap statistic works relatively well under low to moderate contamination levels. However, its performance may degrade when the contamination level is high, especially for the soft-thresholding-based methods. The SCAD-thresholding-based methods generally show better robustness and accuracy in identifying outliers and informative variables under various contamination levels.

### 3.3 Simulation 2: comparison with other methods

In this subsection, we present the findings of a comprehensive simulation study conducted to investigate the cluster data structure and properties by the ARSK algorithm based on soft-thresholding and SCAD-thresholding. The study extensively applied Gap statistics to estimate optimal parameter settings and compared with several benchmark approaches, including the original $K$-means (KC), PCA-$K$-means (PCA-KC) - a mixed approach that combines Principal Component Analysis (PCA) with the original $K$-means algorithm, as initially proposed by Ding & He (2004). It is noted that PCA is a widely recognized dimensionality reduction technique that transforms high-dimensional data into a lower-dimensional space. We also considered Trimmed $K$-means (TKM), which executes standard $K$-means clustering and subsequently eliminates a predetermined percentage of points exhibiting the greatest Euclidean distance from their respective cluster centroids. Furthermore, we employ Robust and Sparse $K$-means (RSKC), whose fundamental premise involves adapting the SK-means algorithm of Witten & Tibshirani (2010) in conjunction with Trimmed $K$-means. Lastly, we apply the Weighted Robust $K$-means (WRCK) method, which shares similarities with RSKC but distinguishes itself through the incorporation of observation-specific weights in its objective function. These weights are reflective of each observation's isolation level relative to its surrounding neighbourhood.

Considering the mixture error model mentioned at the beginning, we also generated data for 3 clusters, each containing 50 observations. We set the number of variables as $p = 50$, with $q = 5$, and $p = 500$, with $q = 50$. Meanwhile, we consider $\pi$ in $\{0, 0.1, 0.2\}$ for specifying the distribution of outliers. As previously stated, we study two types of covariance matrices: one with $\boldsymbol{\Sigma}_p = \boldsymbol{I}_p$ and the another generated according to the method proposed by Hirose et al. (2017) as

$$\boldsymbol{\Sigma}_p = \boldsymbol{Q} \begin{bmatrix} 1 & \rho_t & \cdots & \rho_t \\ \rho_t & 1 & \ddots & \rho_t \\ \vdots & \ddots & 1 & \vdots \\ \rho_t & \cdots & \rho_t & 1 \end{bmatrix} \boldsymbol{Q}^T,$$

where the $\boldsymbol{Q}$ denotes a $p \times p$ random rotation matrix satisfying $\boldsymbol{Q}^T = \boldsymbol{Q}^{-1}$ and the $\rho_t$ are randomly generated form an Uniform distribution $U(0.1, 1)$.

In order to evaluate the clustering accuracy and outlier detection capability of each approach, we employ the clustering error rate (CER) as described above. To assess the outlier detection performance, the outliers identified by each model are assigned to the $(K + 1)$-th cluster group. Table 3.3 presents the results of 100 simulations for the scenario where all variables are independent, i.e., $\boldsymbol{\Sigma}_p = \boldsymbol{I}_p$, while Table 3.3 summarizes the results of 100 simulations for the scenario where variables exhibit a certain correlation structure.

In Tables 3.3 and 3.3, the results show that KC, PCA-KC, and TKM approaches, which lack robustness for high dimensions or outliers, are the worst performers, particularly when the dimensionality $p$ and proportion of outliers $\pi$ increase. For instance, in the independent case with $p = 500$ and $q = 50$, the CER of KC increases from 0.050 to 0.341 as $\pi$ increases from 0 to 0.2. Similarly, the CER of PCA-KC and TKM

| | p = 50 | | | p = 500 | | |
| | q = 5 | | | q = 50 | | |
| | $\pi = 0$ | $\pi = 0.1$ | $\pi = 0.2$ | $\pi = 0$ | $\pi = 0.1$ | $\pi = 0.2$ |
|---|---|---|---|---|---|---|
| KC | 0.073 (0.119) | 0.191 (0.124) | 0.285 (0.107) | 0.050 (0.105) | 0.228 (0.154) | 0.341 (0.145) |
| PCA-KC | 0.109 (0.113) | 0.326 (0.167) | 0.383 (0.132) | 0.058 (0.113) | 0.219(0.098) | 0.300(0.123) |
| TKM ($\alpha = 0.1$) | 0.140 (0.027) | 0.093 (0.036) | 0.220 (0.087) | 0.128 (0.005) | 0.098 (0.044) | 0.302 (0.146) |
| RSKC ($\alpha = 0.1$) | 0.119 (0.029) | 0.008 (0.031) | 0.169 (0.118) | 0.105 (0.007) | 0 (0) | 0.123 (0.129) |
| RSKC ($\alpha = 0.2$) | 0.208 (0.029) | 0.127 (0.015) | 0.005 (0.021) | 0.187 (0.007) | 0.123 (0.009) | 0 (0) |
| WRCK | 0.129 (0.037) | 0.208 (0.104) | 0.136 (0.071) | 0.118 (0.024) | 0.084 (0.028) | 0.046 (0.018) |
| soft-soft-ARSK | 0.010 (0.043) | 0.014 (0.043) | 0.032 (0.064) | 0 (0) | 0 (0) | 0 (0) |
| soft-SCAD-ARSK | 0.010 (0.038) | 0.013 (0.035) | 0.026 (0.033) | 0 (0) | 0 (0) | 0 (0) |
| SCAD-soft-ARSK | 0.021 (0.059) | 0.016 (0.036) | 0.031 (0.061) | 0 (0) | 0 (0) | 0 (0) |
| SCAD-SCAD-ARSK | 0.017 (0.057) | 0.023 (0.052) | 0.019 (0.045) | 0 (0) | 0 (0) | 0 (0) |

Table 2: When variables are independent, the average values of the CER and their standard errors (in parentheses) based on 100 Monte Carlo replications.

| | p = 50 | | | p = 500 | | |
| | q = 5 | | | q = 50 | | |
| | $\pi = 0$ | $\pi = 0.1$ | $\pi = 0.2$ | $\pi = 0$ | $\pi = 0.1$ | $\pi = 0.2$ |
|---|---|---|---|---|---|---|
| KC | 0.078(0.122) | 0.199(0.123) | 0.318(0.138) | 0.037(0.096) | 0.219(0.172) | 0.306(0.136) |
| PCA-KC | 0.112(0.113) | 0.284(0.169) | 0.372(0.126) | 0.113(0.131) | 0.212(0.075) | 0.278(0.046) |
| TKM ($\alpha = 0.1$) | 0.083(0.055) | 0.039(0.080) | 0.240(0.104) | 0.065(0.002) | 0.139 (0.053) | 0.136 (0.286) |
| RSKC ($\alpha = 0.1$) | 0.082 (0.021) | 0.005 (0.016) | 0.235 (0.128) | 0.082 (0.008) | 0 (0) | 0.071 (0.089) |
| RSKC ($\alpha = 0.2$) | 0.194 (0.021) | 0.123 (0.021) | 0.008 (0.015) | 0.162 (0.010) | 0.122 (0.009) | 0 (0) |
| WRCK | 0.119 (0.021) | 0.203 (0.116) | 0.146 (0.071) | 0.126 (0.019) | 0.102 (0.020) | 0.022 (0.081) |
| soft-soft-ARSK | 0.009 (0.038) | 0.014 (0.039) | 0.037 (0.047) | 0 (0) | 0 (0) | 0.000 (0.002) |
| soft-SCAD-ARSK | 0.005 (0.020) | 0.015 (0.039) | 0.032 (0.026) | 0 (0) | 0 (0) | 0.001 (0.002) |
| SCAD-soft-ARSK | 0.007 (0.021) | 0.009 (0.014) | 0.047 (0.069) | 0 (0) | 0 (0) | 0.000 (0.002) |
| SCAD-SCAD-ARSK | 0.005 (0.019) | 0.004 (0.005) | 0.029 (0.033) | 0 (0) | 0 (0) | 0.002 (0.018) |

Table 3: When variables are correlated, the average values of the CER and their standard errors (in parentheses) based on 100 Monte Carlo replications.

($\alpha = 0.1$) increase from 0.058 to 0.300 and 0.128 to 0.302, respectively, under the same scenario. Conversely, RSCK, WRCK, and ARSK perform exceptionally well in high-dimension and high-contamination scenarios. However, the effectiveness of the RSCK approach heavily depends on the choice of the parameter $\alpha$. When $\alpha$ equals the true outlier proportion, RSCK performs impressively, with CER close to 0. For example, in the correlated case with $p = 500$ and $q = 50$, the CER of RSKC ($\alpha = 0.1$) is 0 when $\pi = 0.1$. However, the approach loses effectiveness if $\alpha$ is incorrectly specified, leading to higher CER values. In the same scenario, the CER of RSKC ($\alpha = 0.2$) is 0.122 when $\pi = 0.1$. The WRCK approach, based on the density clustering method, generally outperforms RSCK when outliers are present. Still, it may erroneously classify more normal data points as outliers when no contamination exists. In contrast, the ARSK approach, with different combinations of thresholding functions (e.g., soft-soft, soft-SCAD, SCAD-soft, and SCAD-SCAD), consistently maintains a low Clustering Error Rate (CER) across all simulated data scenarios.

In summary, the simulation results demonstrate the effectiveness and robustness of the ARSK method in clustering high-dimensional datasets, particularly in the presence of outliers and variable correlations. Across diverse data scenarios, the ARSK variants consistently outperform traditional methods, including KC, PCA-KC, TKM, RSCK, and WRCK, highlighting the distinct advantages of the ARSK approach.

Both In Table 3.3 and Table 3.3, we observe that the CER of KC is worse than ARSK when there are no outliers in the dataset. Moreover, when $\pi = 0.1$, the CER of the TKM method is significantly higher than that of the RSCK method. We attribute this result to the presence of noisy variables, which contribute to the increased CER. Therefore, it is essential to identify these noisy variables to reduce their impact.

Through 100 repetitions, the results of the TPR and TNR are presented in Tables 3.3 and 3.3. As the proportion of outliers increases from 0 to 0.2, both the TPR and TNR of most approaches demonstrate a decreasing trend across different scenarios, suggesting that the presence of outliers can negatively influence the variable selection performance. However, the TPR and TNR of each type of ARSK approach variant remain relatively high and stable across various data scenarios, demonstrating ARSK's superior variable selection capabilities in the presence of outlier contamination in datasets. For example, in the independent case with $p = 500$ and $q = 50$, the TPR of soft-soft-ARSK, soft-SCAD-ARSK, SCAD-soft-ARSK, and SCAD-SCAD-ARSK remain above 0.797, 0.810, 0.798, and 0.810, respectively, even when $\pi = 0.2$. Similarly, their TNR values stay above 0.977, 0.980, 0.978, and 0.981 in the same scenario. These results emphasize the superior ability of soft-ARSK and SCAD-ARSK methods to accurately identify informative and non-informative variables in the presence of outliers, making them valuable tools for handling contaminated clustering datasets.

In contrast, during the simulation process, we observed that traditional trimmed approaches such as WRCK and RSKC often struggled to maintain their variable selection performance under high outlier proportions. In many cases, these methods incorrectly assigned non-zero weights to variables that originally had zero weights, leading to an increased number of false positives. This limitation makes it challenging for WRCK and RSKC to effectively detect informative variables in the presence of outliers, emphasizing the need for more effective methods like ARSK.

| | | $p = 50$ | | | $p = 500$ | | |
| | | $q = 5$ | | | $q = 50$ | | |
| | | $\pi = 0$ | $\pi = 0.1$ | $\pi = 0.2$ | $\pi = 0$ | $\pi = 0.1$ | $\pi = 0.2$ |
|---|---|---|---|---|---|---|---|
| TPR | RSKC ($\alpha = 0.1$) | 0.924 (0.038) | 0.890 (0.100) | 0.710 (0.267) | 0.845 (0.101) | 0.908 (0.100) | 0.780 (0.255) |
| | RSKC ($\alpha = 0.2$) | 0.654 (0.109) | 0.728 (0.143) | 0.880 (0.098) | 0.690 (0.043) | 0.993 (0.027) | 0.917 (0.095) |
| | WRCK | 0.888 (0.099) | 0.664 (0.172) | 0.834 (0.106) | 0.734 (0.044) | 0.978 (0.048) | 0.841 (0.101) |
| | soft-soft-ARSK | 0.922 (0.127) | 0.842 (0.169) | 0.810 (0.183) | 0.961 (0.035) | 0.853 (0.053) | 0.797 (0.084) |
| | soft-SCAD-ARSK | 0.944 (0.103) | 0.910 (0.162) | 0.800 (0.171) | 0.932 (0.020) | 0.836 (0.084) | 0.818 (0.018) |
| | SCAD-soft-ARSK | 0.962 (0.092) | 0.906 (0.153) | 0.766 (0.144) | 0.964 (0.029) | 0.969 (0.024) | 0.798 (0.054) |
| | SCAD-SCAD-ARSK | 0.970 (0.076) | 0.802 (0.150) | 0.810 (0.160) | 0.969 (0.026) | 0.820 (0.054) | 0.810 (0.055) |
| TNR | RSKC ($\alpha = 0.1$) | 0.671 (0.427) | 0.660 (0.475) | 0.926 (0.102) | 0.806 (0.377) | 0.548 (0.490) | 0.817 (0.235) |
| | RSKC ($\alpha = 0.2$) | 0.978 (0.142) | 0.880 (0.326) | 0.753 (0.430) | 1 (0) | 0.086 (0.259) | 0.532 (0.482) |
| | WRCK | 0.660 (0.476) | 0.911 (0.199) | 0.837 (0.309) | 1 (0) | 0.261 (0.416) | 0.536 (0.478) |
| | soft-soft-ARSK | 1 (0) | 0.969 (0.048) | 0.981 (0.042) | 0.999 (0.001) | 0.986 (0.012) | 0.977 (0.009) |
| | soft-SCAD-ARSK | 1 (0) | 0.954 (0.040) | 0.996 (0.008) | 0.972 (0.001) | 0.988 (0.014) | 0.980 (0.009) |
| | SCAD-soft-ARSK | 1 (0) | 0.906 (0.153) | 0.976 (0.021) | 0.999 (0.001) | 0.999 (0.001) | 0.978 (0.008) |
| | SCAD-SCAD-ARSK | 1 (0) | 0.976 (0.021) | 0.969 (0.017) | 0.999 (0.000) | 0.981 (0.010) | 0.981 (0.010) |

Table 4: When variables are independent, the average of 100 tests of TPR and TNP for variable selections and its standard error(in parentheses) are presented for the various data scenarios

## 3.4 Applications to real data

### 3.4.1 UCI data application

In this subsection, we apply ARSK with different thresholding configurations to real-life datasets and compare it with the benchmark methods previously mentioned in the simulation study. We consider the dataset all from the UCI Machine Learning Repository (Dua & Graff (2019)). The datasets evaluated include various applications, such as glass identification, breast cancer diagnosis, acoustic signal classification, spam base detection, mortality analysis, and Parkinson's disease detection. The real-life dataset comprises data with a wide range of dimensions, ranging from low to high. As in the previous analysis, we also applied Gap statistics to estimate the hyperparameter. Due to the different units of the continuous variables, we normalize all of continuous variables before conducting the study.

Table 6 summarizes the result of the real-life dataset comparison. The reported values are CERs for each approach. In order to better illustrate the performance of that method, we have emphasized the first and second best CER values in bold when applied to a certain real-life dataset. While traditional methods such

| | | $p = 50$ | | | $p = 500$ | | |
| | | $q = 5$ | | | $q = 50$ | | |
| | | $\pi = 0$ | $\pi = 0.1$ | $\pi = 0.2$ | $\pi = 0$ | $\pi = 0.1$ | $\pi = 0.2$ |
|---|---|---|---|---|---|---|---|
| TPR | RSKC ($\alpha = 0.1$) | 0.917 (0.013) | 0.839 (0.125) | 0.761 (0.236) | 0.813 (0.125) | 0.934 (0.092) | 0.802 (0.294) |
| | RSKC ($\alpha = 0.2$) | 0.705 (0.132) | 0.608 (0.260) | 0.860 (0.015) | 0.695 (0.232) | 0.903 (0.057) | 0.884 (0.072) |
| | WRCK | 0.813 (0.126) | 0.604 (0.212) | 0.734 (0.126) | 0.613 (0.213) | 0.932 (0.074) | 0.812 (0.126) |
| | soft-soft-ARSK | 0.953 (0.086) | 0.860 (0.175) | 0.769 (0.042) | 0.974 (0.031) | 0.843 (0.071) | 0.733 (0.095) |
| | soft-SCAD-ARSK | 0.940 (0.105) | 0.855 (0.181) | 0.806 (0.177) | 0.968 (0.022) | 0.849 (0.071) | 0.773 (0.109) |
| | SCAD-soft-ARSK | 0.928 (0.096) | 0.830 (0.177) | 0.772 (0.169) | 0.973 (0.029) | 0.877 (0.051) | 0.813 (0.063) |
| | SCAD-SCAD-ARSK | 0.928 (0.104) | 0.918 (0.120) | 0.758 (0.186) | 0.968 (0.036) | 0.874 (0.053) | 0.793 (0.058) |
| TNR | RSKC ($\alpha = 0.1$) | 0.680 (0.387) | 0.730 (0.325) | 0.864 (0.226) | 0.776 (0.218) | 0.632 (0.363) | 0.743 (0.206) |
| | RSKC ($\alpha = 0.2$) | 0.837 (0.232) | 0.935 (0.102) | 0.794 (0.400) | 1 (0) | 0.136 (0.454) | 0.542 (0.436) |
| | WRCK | 0.697 (0.437) | 0.954 (0.083) | 0.803 (0.231) | 1 (0) | 0.476 (0.376) | 0.644 (0.306) |
| | soft-soft-ARSK | 1 (0) | 0.961 (0.085) | 0.985 (0.026) | 0.998 (0.006) | 0.977 (0.011) | 0.984 (0.015) |
| | soft-SCAD-ARSK | 1 (0) | 0.876 (0.080) | 0.997 (0.008) | 0.999 (0.001) | 0.978 (0.012) | 0.991 (0.008) |
| | SCAD-soft-ARSK | 1 (0) | 0.978 (0.035) | 0.992 (0.017) | 0.983 (0.010) | 0.977 (0.011) | 0.983 (0.010) |
| | SCAD-SCAD-ARSK | 1 (0) | 0.968 (0.020) | 0.994 (0.013) | 0.997 (0.005) | 0.986 (0.010) | 0.981 (0.009) |

Table 5: When variables are correlated, the average of 100 tests of TPR and TNP for variable selections and its standard error(in parentheses) are presented for the various data scenarios

| | | | | | | RSKC | | | ARSK (proposed) | | | |
| dataset | $K$ | $(n, p)$ | KC | PCA-KC | TKM | $\alpha = 0.1$ | $\alpha = 0.2$ | WRCK | soft-soft | soft-SCAD | SCAD-soft | SCAD-SCAD |
|---|---|---|---|---|---|---|---|---|---|---|---|---|
| glass | 7 | $(214, 9)$ | 0.334 | 0.299 | 0.324 | 0.324 | 0.304 | 0.319 | 0.259 | 0.297 | **0.159** | **0.142** |
| Breast Cancer | 2 | $(699, 9)$ | 0.080 | 0.080 | 0.107 | 0.071 | 0.081 | 0.104 | **0.065** | **0.057** | 0.079 | 0.084 |
| Acoustic | 4 | $(400, 50)$ | 0.309 | 0.443 | 0.392 | **0.293** | **0.280** | 0.382 | 0.354 | 0.360 | 0.355 | 0.346 |
| Spambase | 2 | $(4601, 57)$ | 0.476 | 0.481 | **0.310** | 0.406 | 0.563 | 0.493 | 0.434 | 0.432 | **0.364** | 0.413 |
| mortality | 3 | $(198, 90)$ | 0.227 | 0.185 | 0.228 | 0.153 | 0.149 | **0.106** | **0.131** | 0.136 | 0.143 | 0.149 |
| DARWIN | 2 | $(174, 451)$ | 0.391 | 0.400 | 0.447 | 0.402 | 0.434 | 0.465 | **0.385** | 0.398 | 0.454 | 0.401 |
| Parkinson | 2 | $(756, 754)$ | 0.477 | 0.462 | 0.500 | 0.482 | 0.563 | **0.392** | 0.445 | 0.441 | **0.429** | 0.462 |

Table 6: Comparison of CERs of different algorithms for the real-world data. The smallest and second smallest CER values are shown in bold.

as KC, PCA-KC, and TKM demonstrate effectiveness in certain contexts, they generally yield higher CERs across the datasets compared to the proposed ARSK algorithm and its variants. For instance, in the glass and Breast Cancer dataset, the CER values obtained by KC, PCA-KC, and TKM are considerably higher than those of the best-performing ARSK. In seven real-world datasets, our ARSK method ranked among the top two best methods in six of the datasets. Therefore, the proposed ARSK algorithm consistently achieves superior performance with lower CERs across most of the real datasets, particularly when using the multivariate soft-threshold operator to obtain the error matrix $\boldsymbol{E}$. ARSK demonstrates competitive performance in this regard. Furthermore, Figure 3.4.1 summarizes the number of zero elements in the variable weight vector of each method, where ARKC adopted a combination of thresholding corresponding to the minimum CER. Figure 3.4.1 shows that when analyzing ultra-high-dimensional data, the variable weight vector predicted by ARSK exhibits strong sparsity.

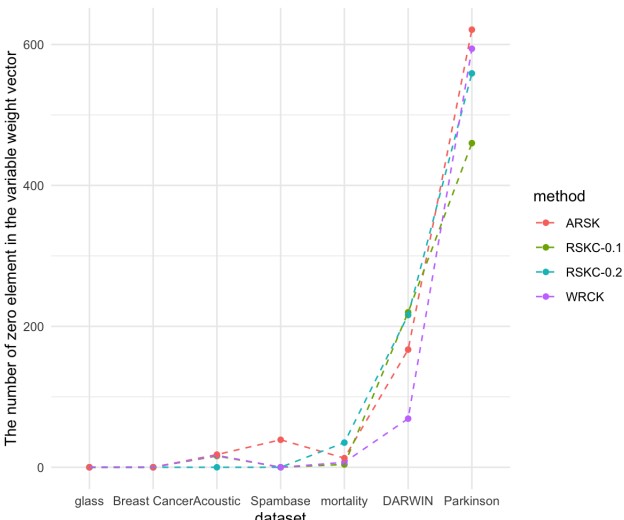

Figure 1: The number of zero elements in the variable weight vector of each method, with results ranging from low-dimension to high-dimension dataset

This comparative analysis emphasizes the ARSK algorithm's success in significantly reducing classification error rates across diverse real-world datasets, showcasing the versatility and adaptability of our proposed clustering method. Its different configurations (e.g., soft-soft, soft-SCAD, SCAD-soft, and SCAD-SCAD) allow it to handle various data structures and sizes effectively. The consistent performance across different real-world datasets underscores the broad applicability of our approach in various fields.

### 3.4.2 Analyzing the group structure of recognition of handwritten digits

The dataset was downloaded from Kaggle, a data science competition platform. It consists of digitized images of the numbers 0 to 9 handwritten by 13 subjects. The images were divided into 64 non-overlapping blocks of $4 \times 4$ bits. For each block, we observe the number of black pixels. Hence, each image is represented by $p = 64$ variables recording the count of pixels in each block, taking integer values between 0 and 16. Its row names identify the true digit in the image, and the column names identify the position of each block in the original bitmap.

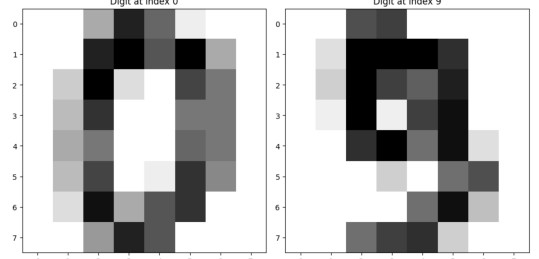

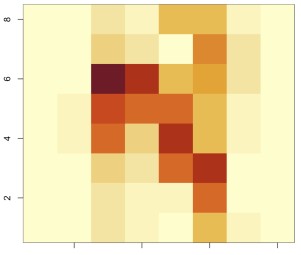

Figure 2: The images of handwritten digits data

Figure 3: The heatmap of the variable weight vector obtained by SCAD-soft ARSK

Our focus is to detect an entire variable weight and group structure, i.e., ten groups. Given the superior performance of SCAD-soft ARSK in subsection 3.4.1, we apply RSKC ($\alpha = 0.1$), RSKC ($\alpha = 0.2$), WRCK, and SCAD-soft ARSK to this dataset. Although the precise contribution of individual pixels to the clustering process remains indeterminate, visual inspection of the real images of handwritten digits data (0 and 9) as Figure 2 shows that the pixels situated in both the initial and terminal columns exhibit uniform white

|       | RSKC-0.2 | RSKC-0.1 | SCAD-soft ARSK | WRCK  |
|-------|----------|----------|----------------|-------|
| CER   | 0.086    | 0.077    | 0.079          | 0.156 |

Table 7: Evaluation of the clustering performance on recognition of handwritten digits

|       | RSKC-0.2 | RSKC-0.1 | SCAD-soft ARSK | WRCK |
|-------|----------|----------|----------------|------|
| CER   | 6        | 9        | 13             | 0    |

Table 8: The number of zero elements within the weight vector derived from each method

colouration. Thus, these 16 pixels can be reasonably deemed redundant and inconsequential for the clustering. The optimal robustness and sparsity parameter for ARSK is also selected using the Gap statistic. The evaluation of the resulting clustering solution is presented in Table 3.4.2. We also examine the final variable weights obtained by the sparse $k$-means-based algorithms. Table 3.4.2 illustrates each method's number of zero elements within the weight vector derived. The analysis of the weight values reveals that WRCK is ineffective in achieving sparsity within the variable weight vector, as evidenced by the exclusive presence of nonzero elements throughout. Correspondingly, it also exhibits the highest CER. As shown in Table 3.4.2, the performance of RSKC and ARSK is similar, with CER values around 0.8. However, the RSKC ($\alpha = 0.2$) produces weight vectors with only 6 zero elements, indicating a moderate sparsity level. In contrast, the proposed ARSK approach generates weight vectors with 13 zero elements that are close to 16. We also presented a heatmap, in Figure 3, visualization of the variable weight vector derived from the SCAD-soft ARSK algorithm as Figure 2, wherein the intensity of the chromatic scale correlates positively with the magnitude of the weights. We can conclude that these findings correspond with those shown in Figure 2. These results demonstrate that ARSK exhibits both sparsity and accuracy of feature selection capabilities.

## 4 Concluding remarks

In this paper, we propose an Adaptively Robust and Sparse $K$-means Clustering algorithm designed to handle multiple tasks within a unified framework for a comprehensive analysis of contaminated high-dimensional datasets. Our approach demonstrates greater flexibility in handling outliers within real-world datasets. We also proposed a modified Gap statistic with an alternating optimization algorithm to search for tuning parameters. This approach significantly reduces the computational time required for parameter tuning compared to traditional methods. The experimental results demonstrate the significant capabilities of the proposed approach in revealing group structures in high-dimensional data. Our model not only performs well in detecting outliers but also identifies significant variables directly, leading to a more thorough understanding of complex clustering datasets.

In our current study, we assume that the number of clusters is known and consistent with the traditional $K$-means method. Additionally, Chen & Witten (2023) proposed a test for detecting differences in means between each cluster estimated from $K$-means clustering. This technique can be applied to the modified dataset $X_{ij} - E_{ij}$, where noise variables (i.e., $w_j = 0$) are removed to help determine which clusters should be retained.

Furthermore, we have consistently utilized a single penalty for outliers in the objective function (4). This implements a uniform approach to detecting outliers across various groups. However, given the complexity and diversity of the data, a more sophisticated strategy may be required. This might involve using distinct group penalty functions for different groups. Incorporating this approach into unsupervised learning presents challenges in selecting group penalty functions. These extensions are left to our future work.

### Acknowledgments

This work is supported by the Japan Society of the Promotion of Science (JSPS KAKENHI) grant numbers, 20H00080 and 21H00699.

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

# A    Appendix: Derivation of the solution to (9)

Consider maximizing the following regularized objective function for estimating the weights $w_j$.

$$L(w_j, \lambda_2) = \sum_{j=1}^{p} w_j Q_j - \sum_{j=1}^{p} \left( P_2(w_j; \lambda_2) + \frac{1}{2} w_j^2 \right),$$

where $Q_j = Q_j^R(\boldsymbol{X}_{:,j}; \Theta, \boldsymbol{E}_{:,j})$. Taking the derivative with respect to $w_j$ and setting it to zero, we obtain:

$$Q_j - w_j - \Gamma_j = 0.$$

When the $L_1$ penalty $P_2(w_j; \lambda_2)$ is applied, $\Gamma_j$ represents the subgradient of the $L_1$ penalty with respect to $w_j$. Solving the above equation yields

$$S^{\text{soft}}(x; \lambda_2) = w_j = \begin{cases} Q_j - \lambda_2, & \text{if } Q_j > \lambda_2 \\ 0, & \text{if } |Q_j| \leq \lambda_2 \\ Q_j + \lambda_2, & \text{if } Q_j < -\lambda_2. \end{cases}$$

When the SCAD penalty $P_2(w_j; \lambda_2)$ is applied, $\Gamma_j$ represents the subgradient of the SCAD penalty with respect to $w_j$. Solving the above equation yields

$$S^{\text{SCAD}}(x; \lambda_2) = w_j = \begin{cases} \text{sgn}(Q_j)(|Q_j| - \lambda_2)_+ & \text{if } |Q_j| \leq 2\lambda_2 \\ (a-2)^{-1}\{(a-1)Q_j - a\lambda_2\text{sign}(Q_j)\} & \text{if } 2\lambda_2 \leq |Q_j| < a\lambda_2 \\ Q_j & \text{if } a\lambda_2 \leq |Q_j|. \end{cases}$$

Projecting $\hat{w}_j$ onto the parameter space $\sqrt{\sum_{j=1}^{p} w_j} = 1$, the final solution can be rewritten as

$$w_j = \frac{S(Q_j; \lambda_2)}{\sqrt{\sum_{j=1}^{p}(S(Q_j; \lambda_2))^2}}.$$

