# OpenReview forum: "Adaptively Robust and Sparse $K$-means Clustering"
_TMLR — Accepted by TMLR_

### Review · Reviewer_S4PT · 2024-08-15

**Summary Of Contributions:**

This paper presents a novel adaptively robust and sparse K-means clustering (ARSK) algorithm, which overcomes the limitations of the conventional K-means algorithm in handling outliers and high-dimensional noisy data. ARSK can improve the robustness of clustering outcomes by incorporating a redundant error component and a group sparse penalty. Furthermore, ARSK reduces the computational time required for parameter tuning compared to traditional methods.

**Audience:**

Yes

**Claims And Evidence:**

Yes

**Requested Changes:**

Please refer to the Weaknesses.

**Strengths And Weaknesses:**

Strengths:
1. The proposed algorithm combines robustness and sparsity, enhancing its resistance to outliers and noise, thereby improving the stability and reliability of clustering results. The algorithm is supported by theoretical.
2. It effectively reduces computational costs and enhances practical utility, with the source code being readily available.

Weaknesses:
1. The use of datasets from the UCI Machine Learning Repository may not fully represent the diversity and complexity of real-world data.
2. The recent comparison algorithms are published in 2016 and 2019, which might not provide a comprehensive view of the current state-of-the-art in clustering algorithms.
3. The comparison algorithms used in the paper lack references to their original papers and necessary explanations. This makes it difficult for readers to understand these algorithms.

---

> ### Author Response · Authors · 2024-09-08
> **Authors’ response to Reviewer S4PT**
>
> Thank you for your kind comment. We have addressed all the issues you listed. Please see our responses below.
>
> **1. The use of datasets from the UCI Machine Learning Repository may not fully represent the diversity and complexity of real-world data.**
>
>   - In addition, we can include analysis of the recognition of handwritten digits from Kaggle, a data science competition platform, in the real-world data analysis section. We have already done analysis and found that we get a similar result for the proposed method.
>
> **2. The recent comparison algorithms were published in 2016 and 2019, which might not provide a comprehensive view of the current state-of-the-art clustering algorithms.**
>
>    - In this study, we focus on robust sparse clustering methods based on the K-means method. Through a comprehensive literature review, we have found that there is relatively limited research in this area. Consequently, our research aims to provide new support in this research field.
>
> **3. The comparison algorithms used in the paper lack references to their original papers and necessary explanations. This makes it difficult for readers to understand these algorithms.**
>
>  - We will provide a more in-depth explanation of the comparison algorithms in the simulation section.

---

> > ### Comment · Reviewer_S4PT · 2024-09-09
> >
> > Thank you for your response. I believe submitting a revision would make the response more convincing. Although the authors did not have sufficient time to prepare a revision, further results and discussion for issues 1 and 3 are essential to improve the clarity and impact of the response.

---

> > > ### Author Response · Authors · 2024-09-10
> > > **Reply to Reviewer S4PT**
> > >
> > > we will upload the revised manuscript as quickly as possible

---

> > > > ### Comment · Reviewer_S4PT · 2024-09-23
> > > >
> > > > In the revised manuscript, the concerns regarding issues 1 and 3 have been largely resolved.

---

> > > > > ### Author Response · Authors · 2024-09-23
> > > > > **Reply to Reviewer S4PT**
> > > > >
> > > > > Thank you for your review, which has helped enhance our manuscript.

---

### Review · Reviewer_n9Js · 2024-08-17

**Summary Of Contributions:**

This paper proposes a k-means formulation for clustering data points while detecting outliers and unimportant features. The proposed algorithm is called adaptively robust and sparse K-means (ARSK). The basis for their work is the algorithm proposed by Witten & Tibshirani (2010) that rephrases the k-means objective function in terms of the "between-cluster sum of squares." In that paper, the goal was to perform variable selection with k-means. In this paper, the authors go one step further and propose a formulation that is also robust to outliers. The main is to estimate a new parameter `E_{ij}` for each observation `x_{ij}`, where `i` is the index of the observation and `j` is the coordinate. `E_{ij}` will contribute to the objective function when `x_{ij}` is an outlier. The authors propose enforcing sparsity to the matrix `E` so that the optimization can't "mark" all points as outliers. The algorithm is validated on synthetic experiments and over UCI datasets.

**Audience:**

Yes

**Claims And Evidence:**

Yes

**Requested Changes:**

My main recommendation is a text revision. There are couple of typos that a minor revision would fix it, for example, capital letters in the middle of the text and phrases that could be shorter and clearer.

**Strengths And Weaknesses:**

*Strength*: I think the main idea is interesting and relatively easy-to-follow. If my interpretation is correct, that's a neat solution to find outliers.

*Weakness*: Text quality could be improved. Specially, the algorithm's presentation and motivation could be more elaborated. Instead of motivating the problem, the introduction reads like a "related work" section, which sets your formulation as an incremental combination of the previous idea. The introduction would be stronger if the authors referred to practical problems that now can be solved by the proposed formulation and were not possible before.

---

> ### Author Response · Authors · 2024-09-08
> **Review of Paper3073 by Reviewer n9Js**
>
> Thank you to all the reviewers and the action editor for your acknowledgment and feedback.
>
> **1. My main recommendation is a text revision. There are couple of typos that a minor revision would fix it, for example, capital letters in the middle of the text and phrases that could be shorter and clearer.**
>  - We carefully reviewed and corrected any spelling errors and unnecessary capitalization issues in the text. Additionally, We will strive to make the sentences more concise and clear to enhance the readability of the article. After incorporating all the revisions suggested by the three reviewers, we will submit an updated manuscript.
> We have also refined the introduction section to highlight the superior performance of our method on high-dimensional and heavily contaminated datasets, in comparison to previous research. Unlike previous methods that solely apply $L_1$ penalties to variable weights, our approach emphasizes flexibility by allowing the free choice of penalty functions within the model.

---

### Review · Reviewer_BVFi · 2024-09-03

**Summary Of Contributions:**

- This work proposes a $K$-means clustering algorithm.
- It proposes adaptively robust and sparse $K$-means clustering.
- The authors introduce an error component for each observation to enhance robustness.
- Several experimental results are demonstrated to show the effectiveness of the proposed method.
- In addition, source code has been provided in the submission.

**Audience:**

Yes

**Broader Impact Concerns:**

I don't have particular concerns on the broader impact of this work, but the authors might be able to discuss them.

**Claims And Evidence:**

Yes

**Requested Changes:**

- What is SCAD? It is not defined in the submission.
- I think the notation of $x_{i, :}$ is too complex. Why don't you simply use $x_{i}$?
- $(x\_{i, :} - x\_{i', :})^2$ should be the squared value of $L_2$-distance.
- Could you provide the computational complexity of the iterative algorithm, which is described in Algorithm 1?
- I think that Sections 2.4 and 3.2 are useful for understanding how to tune $\lambda_1$ and $\lambda_2$, but could you provide a practical way to tune them?
- Could you add some visual example of your algorithm that can show the strengths of the proposed algorithm? It will be helpful to efficiently understand your method.
- What is $E_{\cdot j}$ in the last paragraph of Page 4?

**Strengths And Weaknesses:**

### Strengths

- It solves a well-defined problem on adaptively robust and sparse clustering.
- A paper is generally well-written.
- The proposed method is analyzed appropriately.

### Weaknesses

- Presentation can be improved more.

---

> ### Author Response · Authors · 2024-09-08
> **Review of Paper3073 by Reviewer BVFi**
>
> We thank the reviewer for the appreciation of our work.
>
> **1. What is SCAD? It is not defined in the submission.**
>  - The Smoothly Clipped Absolute Deviation (SCAD) penalty is a regularization technique that shares similarities with the LASSO method. However, compared to LASSO, solutions obtained using the SCAD penalty tend to be more sparse
>
> **2. I think the notation of $x_{i, :}$ is too complex. Why don't you simply use $x_{i}$?**
>  - First, please note that $x_{i, :}$ is a vector and its element is denoted by $x_{ij}$. Hence, if we use $x_i$ instead of $x_{i, :}$, there may be confusion with $x_{ij}$.Also, $x_{i, :}$ can be regarded as $i$th vector of the data matrix. According to the guideline of TMLR, we need to use the notation $x_{i, :}$ to express $i$th row vector of the data matrix.
>
> **3. $(x_{i, :} - x_{i', :})^2$ should be the squared value of $L_2$-distance**
>  - Thank you for pointing this out. We have corrected it.
>
> **4. Could you provide the computational complexity of the iterative algorithm, which is described in Algorithm 1?**
>  - Assume that the number of iterations in the 4th step is $O(t)$, the computation complexity of $T$ iteration of our algorithm is $O(nptTK)$.We will declare this issue in the revised manuscript.
>
> **5. I think that Sections 2.4 and 3.2 are useful for understanding how to tune $\lambda_1$ and $\lambda_2$, but could you provide a practical way to tune them?**
>  - In Algorithm 2, we have already provided detailed steps for tuning $\lambda_1$ and $\lambda_2$.
>
> **6. Could you add some visual example of your algorithm that can show the strengths of the proposed algorithm? It will be helpful to efficiently understand your method.**
>  - We decided to make a figure summarizing the classification error rate (CER) and the number of features used in clustering for comparative methods (including the proposed one), which would show that the proposed method can achieve high performance of classification with use of the small number of features.
>
> **7. What is $E_{\cdot j}$ in the last paragraph of Page 4?**
>  - $E_{\cdot j}$ represents the $j$-th column of an $n \times p$ error matrix $E$.However, according to the guideline of TMLR, $E_{\cdot j}$ should be expressed as $E_{:, j}$.We also have corrected it.

---

> > ### Comment · Reviewer_BVFi · 2024-09-08
> >
> > The authors did not submit a revision.  I believe that the TMLR review process requires submitting a revision in order to accommodate reviewers' requests.
> >
> > For the response on SCAD, I am sorry for confusing the authors. I would say that SCAD is not defined in the paper or at least the citation of SCAD should be provided.  Also, the full name of SCAD is not described.

---

> > > ### Comment · Action_Editor_srZN · 2024-09-08
> > > **Clarification**
> > >
> > > To clarify, the [TMLR guidelines](https://jmlr.csail.mit.edu/tmlr/editorial-policies.html) do not *oblige* the authors to post a revision of the manuscript:
> > >
> > > > 3. **Rebuttal and discussion**. Reviews will be visible to the authors as they are submitted, but the reviews will not be visible to the public nor to the other reviewers until all the reviews are submitted in order to keep them independent. **Authors may post rebuttals and update their papers in response to the reviews**, and the reviewers and editor may privately discuss the paper.* Authors can respond to a review as soon as it is posted, however we recommend waiting until all 3 reviews have been submitted before submitting any revised version of the PDF manuscript. Reviewers will be able to submit their final recommendations once at least two weeks have elapsed after all 3 reviews became public.
> > >
> > > However, reviewers are encouraged to give constructive feedback that they believe can improve the manuscript, and authors are expected to discuss with the reviewers in case they disagree with specific suggestions. In particular, reviewers can take this interaction (or lack of thereof) into account in their final recommendation.

---

> > > > ### Comment · Reviewer_BVFi · 2024-09-08
> > > >
> > > > Thank you for clarifications.
> > > >
> > > > Nevertheless, I think that the submission of a revision might be helpful to resolve my concerns.

---

> > > > > ### Author Response · Authors · 2024-09-10
> > > > > **Reply to Reviewer BVFi**
> > > > >
> > > > > Sure, we will upload the revised manuscript as quickly as possible

---

> > > > > > ### Comment · Reviewer_BVFi · 2024-09-23
> > > > > >
> > > > > > Thank you for your revised manuscript.
> > > > > >
> > > > > > I think my concerns are mostly resolved.
> > > > > >
> > > > > > One thing you should fix is that a figure caption should be put below a figure.  Please fix it.

---

> > > > > > > ### Author Response · Authors · 2024-09-23
> > > > > > > **Reply to Reviewer BVFi**
> > > > > > >
> > > > > > > We sincerely appreciate your review.
> > > > > > > We will address the figure caption issue in the final manuscript.

---

### Author Response · Authors · 2024-09-18
**Update of manuscript**

Dear Reviewers,

We have completed the manuscript revisions based on the reviewer's suggestions. The blue parts in the text represent the modified sections.
 -We have analyzed handwritten digit data, and a pixel weight heatmap for the handwritten digit data has been added to make the results more intuitive.
 -We have included Figure 1, which shows the number of zero elements in the variable weight vector for each method when handling datasets of different dimensions.
 -The time complexity of the algorithms has been added. Further explanations have been provided for the benchmark methods.
 -We have corrected typo errors.
 -We have put the figure 1 caption below the figure.
Please let us know if you have any concerns.

---

### Decision · Action_Editor_srZN · 2024-10-15

**Recommendation:** Accept with minor revision

**Comment:**

All reviewers agree the method proposed in this work is of interest and deserve publication at TMLR. However, they were also unanymous in pointing out that the paper would benefit from a throughout revision. During the discussion period, the reviewers made some suggestions that the authors took into account.

However, in their recommendation reviewer *n9Js* pointed out that the text is still confusing, giving the following examples:

> *"Moreover, if the approach learns to handle unusual data by adjusting outliers instead of removing them, it particularly helps to preserve the continuity of estimating parameters"*

> *"Meanwhile, to reduce the dimension of data, we also introduce a SCAD penalty to weights for each variable to eliminate noise variables irrelevant to clustering. Based on our analysis of real-world datasets, in the ultra-high dimensional setting (i.e., p > 700), the result procured through the implementation of the lasso penalty proposed by Witten & Tibshirani (2010) frequently poor performance. Therefore, we have developed a methodology based on the SCAD penalty. The strong sparsity property of the SCAD penalty effectively improves the sparsity of estimating the variable weight vector in the ultra-high dimensional scenario."*

- p is only defined later.
- donated -> denoted

> *"A proper of hyperparameters $\lambda_{1}$ and $\lambda_{2}$ should enable the model to accurately identify the number of outliers and the number of informative variables in different contamination levels for"*

Following the reviewers recommendation, I am recommending the manuscript for publication to TMLR. However, I do urge the authors to make a throughout revision of the text, including a spelling check to improve readability.

**Audience:**

All reviewers agree that the results in this work is of interest to the TMLR audience.

**Claims And Evidence:**

This work introduces a novel algorithm, ARSK, designed to enhance the robustness and sparsity of the traditional K-means clustering algorithm. ARSK addresses two key limitations: sensitivity to outliers and inefficiency in handling high-dimensional noisy data. The method incorporates an error component for each observation to manage outliers and applies a group sparse penalty to promote robustness. Additionally, it implements a penalty on variable weights to handle noisy dimensions, leading to sparse clustering. Tuning parameters for controlling robustness and sparsity are selected using Gap statistics. Experimental results demonstrate that ARSK outperforms existing methods by identifying clusters accurately in the presence of both outliers and irrelevant variables.

---

> ### Author Response · Authors · 2024-11-03
> **Official Comment by Authors**
>
> We appreciate the comprehensive efforts of AE in the submission and your expertise and judgment.
> We have modified it according to AE's suggestions in the text and carefully reviewed it. We also revised the entire manuscript including correcting spelling mistakes, clarifying confusing sentences, and adjusting some mathematical symbols. We have uploaded the revised version.